# Exploring the Quantitative Assessment of Spatial Risk in Response to Major Epidemic Disasters in Megacities: A Case Study of Qingdao

**DOI:** 10.3390/ijerph20043274

**Published:** 2023-02-13

**Authors:** Qimeng Ren, Ming Sun

**Affiliations:** School of Landscape, Northeast Forestry University, Harbin 150040, China

**Keywords:** major epidemic diseases, quantitative disaster risk assessment, SDNA, GIS, POI

## Abstract

With the global spread of various human-to-human epidemics, public health issues have become a focus of attention. Therefore, it is of great importance to improve the quantitative risk assessment of the construction of resilient cities in terms of epidemic disasters. Starting with the dimensions of social activities and material space, this paper took Qingdao, China, with a population of 5 million, as an example, and took its seven municipal districts as the research scope. In this paper, five risk factors, including the Population density index, Night light index, Closeness index of roads, Betweenness index of roads and Functional mixed nuclear density index were selected for weighted superposition analysis. We conducted a quantitative assessment of the spatial risk of epidemic disaster so as to obtain the classification and spatial structure of the epidemic disaster risk intensity. The results show that: ① The roads with a large traffic flow are most likely to lead to the risk of urban spatial agglomeration, and the areas with a large population density and large mixture of infrastructure functions are also important factors causing the risk of epidemic agglomeration. ② The analysis results regarding the population, commerce, public services, transportation, residence, industry, green space and other functional places can reflect the high-risk areas for epidemic diseases with different natures of transmission. ③ The risk intensity of epidemic disasters is divided into five risk grade areas. Among them, the spatial structure of epidemic disasters, composed of the first-level risk areas, is characterized by “one main area, four secondary areas, one belt and multiple points” and has the characteristics of spatial diffusion. ④ Catering, shopping, life services, hospitals, schools and transportation functional places are more likely to cause crowd gathering. The management of these places should be focused on prevention and control. At the same time, medical facilities should be established at fixed points in all high-risk areas to ensure the full coverage of services. In general, the quantitative assessment of the spatial risk of major epidemic disasters improves the disaster risk assessment system in the construction of resilient cities. It also focuses on risk assessment for public health events. It is helpful to accurately locate the agglomeration risk areas and epidemic transmission paths that are prone to outbreak or cause epidemic transmission in cities so as to assist the relevant practitioners in containing the epidemic from the initial stage of transmission in a timely manner and prevent the further spread of the epidemic.

## 1. Introduction

In recent years, major outbreaks of human-to-human transmission, such as COVID-19, SARS and influenza A (H1N1), have had significant impacts on urban development and human life in various countries. Dealing with pandemic disasters is one of the major health risk challenges of the 21st century. Interpersonal infectious diseases are divided into respiratory tract transmission, digestive tract transmission, contact transmission, blood transmission, insect-borne transmission, mother-to-child transmission and sexual transmission. These diseases have strong characteristics of transmission from person to person. As dynamic and complex systems, modern cities have increasingly become risk centers of various disasters due to their population concentration, complex functions, developed transportation and dense buildings. These factors also aggravate the generation and spread of different types of infectious diseases.

Canadian ecologist Hollin proposed a resilient city theory characterized by dynamic balance and adaptability. It mainly emphasizes that urban society should comprehensively evaluate the causes, scale and impact of disaster risks when facing various kinds of risk disasters and take corresponding measures to effectively deal with the impacts of risks and curb the further spread and deterioration of risks in a timely manner [1]. Qingdao, which is studied in this paper, is a typical case of a resilient city. Urban disaster risk assessment is an important part of resilient city construction. However, to date, most previous research has focused on natural disasters such as mountain torrents, earthquakes, hurricanes, and so on [2]. However, the risk assessment of infectious diseases is not perfect. In the context of the continuous occurrence of various human-to-human epidemic disasters, it is crucial to assess the risk intensity of epidemic disasters. We must predict the risk areas that might become the sources of outbreak and spread of new outbreaks in the future and counter the weaknesses of resilient city construction. In this way, we could promote the high-quality development of the city.

The spread of major epidemics depends on human contact. Therefore, in the absence of drug intervention and policy prevention and control, the epidemic’s spatial spread is closely related to the population distribution, built environment, scope of activities and the spatial layout of various spatial impact factors. Spatial research is of great significance for the identification of infectious disease hotspots, the analysis of spatial differentiation characteristics, the analysis of high-risk spatial agglomeration and the analysis of spatial transmission networks [3]. As early as the middle of the 20th century, some scholars proposed using geographical thinking to study epidemic prevention and control. When the cholera epidemic broke out in London in 1854, John Snow proposed marking the location of the dead on a map, which led him to discover that the source of cholera transmission was water pollution. This played an important role in containing the epidemic at that time [4]. In recent years, scholars in the field of geography have carried out a series of research works on the spatial transmission of various infectious diseases and achieved remarkable results. Zheng Linlin et al. conducted a statistical analysis of the case data and incidence rate data of respiratory infectious diseases in various provinces of China. They used the method of data statistics to locate the high-risk areas for infectious disease outbreak [5]. Chai Yanwei et al. used the methods of individual space–time path analysis, activity scenario analysis and risk perception mapping to build a geographical framework for the accurate prevention and control of COVID-19 [6]. Guo Liang et al. improved the infectivity dynamics model (SEIR) in combination with the spatial distribution characteristics of population activities, and this method could identify high-risk areas for COVID-19 epidemic on the street scale [7]. Geography academic circles attach great importance to the study of epidemic transmission, which provides a scientific basis for the prevention and control of epidemics [8]. However, the traditional method for the detection of epidemic risk areas is based on the spatial distribution of epidemic case data to build a visual map, and the analysis process is limited to the spatial distribution of the activities of certain groups. There are few studies on the identification of areas at high risk of epidemic due to the absence of case data, drug intervention and policy prevention and control.

Objective and accurate data and analytical methods are very important for establishing a comprehensive and multi-dimensional quantitative indicator system to evaluate the risk intensity of epidemic disasters [9]. The development of new data environments and the emergence of new technologies and methods provide important opportunities for the measurement and analysis of traditional, unmeasurable data [10]. The current research has found that the risk factors affecting the generation and spread of infectious diseases mainly include socio-economic aspects, physical space aspects and aspects related to the natural meteorological environment [11]. Socio-economic aspects mainly include the external input population, economic income level (GDP), density of the population distribution, proportion of the non-agricultural population, proportion of three industries, medical and health levels, average years in education, unemployment rate, policy control of epidemics, poverty rate and proportions of the population representing different ages [12,13]. The physical space aspects mainly include the current situation of land use (the proportion of cultivated land, forest land, water area and construction land), the density and accessibility of the road network, distribution of various infrastructures and the mixture of land use functions [14,15]. The aspects related to the natural meteorological environment mainly include the average surface temperature, precipitation, relative humidity, atmospheric pollutant concentration, atmospheric pressure, annual haze days, DEM elevation, average wind speed and spatial distance [16]. The current research has mainly used geographic weighted regression models, spatial dynamics models, cluster analysis, spatial panel data models and other methods to explore the impacts of different risk factors on the spread of infectious diseases. Few studies have identified areas with hidden dangers of infectious diseases through these factors. Additionally, most of the studies on the risk factors for infectious diseases only involved a single factor or a certain infectious disease. There is a lack of research on the spatial distribution of incidence rates using multiple factors.

For human-to-human epidemic disasters, the high-risk areas in large cities are mostly densely populated areas and areas that are prone to crowd concentration. It is possible to identify risk areas using spatial risk factors that might increase the intensity of crowd activities. Therefore, this study mainly started from the perspective of the spatial layout and only selected risk factors related to the population and spatial geographical distribution of urban built-up areas in order to build an indicator system for analysis. This paper combined the multi-dimensional quantitative assessment of epidemic disasters with the spatial analysis method of using spatial big data to predict the spatial risk of urban agglomeration [17]. The framework of this article is shown in Figure 1.

This paper takes Qingdao, a typical city with a population of 5 million, as an example, and takes its seven municipal districts as the research scope to build an indicator system of social activities and material space. We selected the population density grid data, night lighting data, OSM traffic road network data and six types of POI infrastructure data, including the commercial function, residential function, public service function, traffic function, green space function and industrial function, to conduct spatial analysis in ArcGIS (ArcGIS is developed by ESRI, RedLands, CA, USA) and then obtained five indicators including the Population density index, Night light index, Closeness index of roads, Betweenness index of roads and Functional mixed nuclear density index. Finally, we carried out a quantitative assessment of the spatial risk of epidemic disasters through weighted superposition and identified the risk intensity zoning and the spatial structure of different types of epidemic disasters. We used spatial big data to carry out urban spatial exploration and analysis, built a potential disaster risk indicator system for epidemics, conducted quantitative assessment of the spatial risk of epidemic disasters, predicted the regional risk intensity level of the epidemic area, and identified the spatial structure of epidemic disasters. This study can provide an effective reference for the emergency prevention and control policies of relevant government departments.

## 2. Materials and Methods

### 2.1. Research Scope

In this study, we selected Qingdao, a large coastal city with a population of more than 5 million. As an important coastal central city in China, a coastal resort tourism city, a national historical and cultural city and an international port city, it is typical. Qingdao is located in the southeast coastal area of Shandong Province, connecting Weifang City in the west, Yantai City in the northeast, Rizhao City in the southwest and the Yellow Sea in the southeast. It is located at 119°30′–121°00′ E and 35°35′–37°09′ N. The administrative area of Qingdao includes seven municipal districts and three county-level cities, including Shinan District, Shibei District, Licang District, Laoshan District, Chengyang District, West Coast New Area and Jimo District, as well as Jiaozhou City, Pingdu City and Laixi City. The scope of this study is 105 streets (towns) in seven municipal districts of Qingdao, covering an area of 5226 square kilometers (Figure 2).

As the main urban districts of Qingdao, the seven municipal districts are the main built-up centers of Qingdao, with a developed road network system, perfect infrastructure, strong economic radiation, large population density and high urban vitality. This can easily lead to the large-scale spread of an epidemic. Therefore, the authors constructed a spatial risk indicator system of epidemic disasters within this scope, analyzed the risk intensity zoning and spatial structure of the epidemic disaster, and tested the results using the coronavirus disease as an example. This provides a typical case reference for the quantitative assessment of the spatial risk of major human-to-human infectious diseases.

### 2.2. Technology Route

Michael Batty, an expert of UCL Smart City, once proposed that the economic and social activities, physical space elements and urban functions have a significant relationship with the intersection and interaction of urban complex networks and flow systems [18]. Therefore, starting with the dimensions of urban social activities and physical spaces, we constructed a quantitative assessment index system of spatial risk to identify the risk intensity zoning and spatial structure of major interpersonal epidemic disasters in the megacity [19]. The research steps were divided into five parts: data selection of the risk index, application of population data in identification of epidemic risk areas, spatial processing of the data, factor-weighted superposition analysis and the evaluation and inspection of the disaster risk grade intensity. The research objectives and concepts are shown in Figure 3, and the technical route is shown in Figure 4.

#### 2.2.1. Data Selection of Risk Index

The risk factor indices were divided into two dimensions: the social activity dimension and physical space dimension (Table 1). The dimension of social activities reflects the intensity of crowd activities. It is a direct representation of the risk intensity. For this paper, we selected the Population density index and Night light data index [20]. For the Population density index, we used the population density grid data provided by Worldpop and revised these data on the basis of the seventh census data for 2020 [21]. We used the random forest algorithm to carry out symmetrical redistribution, and 100 × 100 m grid cells were counted. For the Night light data index, we used the NPP-VIIRS night light data obtained by the National Oceanic and Atmospheric Administration of the United States in 2022, with a spatial resolution of 500 m and a radiation resolution of 12 bit. The WGS1984 coordinate system was adopted, and the obtained data do not eliminate the impacts of fishing boat lights, firelight or other temporary lights [22].

The physical space dimension provides a place for the formation of potential risks. For this paper, we selected the Closeness index of roads, Betweenness index of roads and Functional mixed nuclear density index [23]. For the Closeness index of roads and the Betweenness index of roads, we selected traffic road network data obtained from Openstreetmap (OSM) in 2022, including the basic attribute information such as the road name, type, longitude and latitude and the road length [24]. For the Functional mixed nuclear density index, we selected all the POI data crawled from the Gaode Map Open Platform using Python for the seven municipal districts of Qingdao in 2022 (Figure 5). Through data cleaning, screening and pretreatment, a total of 353143 POI data within the study scope were finally obtained [25]. In order to facilitate the later analysis, all the data were reclassified according to the name attribute of POI [26]. According to the provisions of the Guidelines for City County Land Spatial Planning Zoning and Use Classification (201905), the POI data were divided into 13 sub-categories and 6 functional service categories. These are, respectively, commercial (65.4%), residential (2.6%), public service (14.6%), transportation (6%), green space (0.6%) and industrial (10.8%) (Table 2).

#### 2.2.2. Application of Population Data in Identification of Epidemic Risk Areas

The population density, road network, and various functional infrastructure indicators, such as commerce, residence, public service, transportation, green space, and industry, can reflect the population distribution. For human-to-human epidemic disasters, the high-risk areas in large cities are mostly in densely populated areas and areas that are prone to crowd concentration. Therefore, it is necessary to detect the high-risk areas of population concentration through various spatial layout indicators.

#### 2.2.3. Spatial Processing of Data

The traditional method of dividing basic spatial units by road network causes an uneven parcel scale and shape. This can easily lead to a phenomenon in which much POI is contained in a large area plot and the point density is low. This leads to errors in the subsequent weighted superposition analysis [27]. Therefore, for this paper, we selected a 1000 m × 1000 m-scale regular fishnet to divide the study area, and there were 6021 grids after sorting. Through spatial connection analysis in ArcGIS, the processed population grid data, night light data, closeness and betweenness data of roads after SDNA analysis and the nuclear density analysis data derived from various POI data were connected with the fishnets to establish spatial relations and provide all the data spatial attributes. Among them, the nuclear density grid data of all the types of POI were assigned weights, and then the Functional mixed nuclear density index data were obtained through analysis [28].

#### 2.2.4. Factor-Weighted Superposition Analysis

We used the entropy method to carry out weighted superposition analysis assessing the aggregation risk characteristics of various indicators [29]. The quantitative assessment of the spatial risk was carried out according to the Population density index, Night light index, Closeness index of roads, Betweenness index of roads and Functional mixed nuclear density index.

#### 2.2.5. Evaluation and Inspection of Disaster Risk Grade Intensity

According to the weighted superposition results of the five indicators, we were able to obtain the epidemic disaster risk intensity grade zoning and the spatial structure of major urban epidemics. Finally, we selected the distribution data of high-risk sites for COVID-19 in the seven municipal districts of Qingdao from 21 November to 10 December 2022 to test the results in order to ensure their accuracy [30] (Table 3). There were 154 data points for this place, and they were compiled from the official data released by the Qingdao Municipal Health and Health Commission of Shandong Province. The acquisition of historical statistical data on the COVID-19 epidemic was performed to verify the accuracy of the prediction results for the risk level in this paper. This does not mean that the quantitative assessment of the spatial risk can only be used for the COVID-19 epidemic.

### 2.3. Research Method

#### 2.3.1. SDNA Accessibility Analysis Method

The full name of SDNA is Spatial Design Network Analysis [31]. It is a new spatial analysis model developed from spatial syntax and is used for analysis in ArcGIS. Compared with space syntax, it can better meet the requirements for the analysis of accessibility in urban planning and design [32]. By setting different analytical radii and measurement methods, we were able to analyze the accessibility of the road network and calculate the travel flow potential of street motor vehicles, bicycles, pedestrians and public transport. This research mainly used “the closeness” and “the betweenness” of SDNA to conduct a spatial analysis of the road network within the seven municipal districts of Qingdao. In this way, we were able to obtain areas with high accessibility and roads with a high traffic potential.

Closeness refers to “the reciprocal of the ratio that is between the sum of the shortest distances from a node to all other nodes in the system and the total number of nodes”. It can be used to indicate the accessibility or centrality of a place in the system [33]. The closeness of the roads is in direct proportion to the sum of the paths to all other roads, in inverse proportion to the centrality and topological integration capacity of the road, in inverse proportion to the attraction potential for traffic and in inverse proportion to accessibility. The calculation formula is as follows:(1)$$NQPD\left(x\right)=\sum_{y\in Rx}\frac{W\left(y\right)P\left(y\right)}{d\left(x,y\right)}$$
where *x* and *y* are two sections; *d* (*x*, *y*) is the shortest path length between sections *x* and *y*; *R* is the search radius; *W* (*y*) is the weight of section *y*; and *P* (*y*) is the weight of *y* within the search radius *R*. In discrete spatial analysis, *P* (*y*) is 0 or 1. In continuous spatial analysis, *P* (*y*) ∈ [0, 1] [34].

Betweenness is also called traversal. It is interpreted as the number of times that a segment *x* is crossed by the shortest path between any other two segments *y* and *z* within a specific search radius. It reflects the potential of this section as a road for crossing traffic [35]. The betweenness of a road section is proportional to the traffic flow passing through it. The calculation formula is as follows:(2)$$TPBt\left(x\right)=\sum_{y\in N}\sum_{z\in R_{y}}OD\left(y,z,x\right)\frac{W\left(z\right)P\left(z\right)}{total\;weight\left(y\right)}$$
where *R* is the search radius; *OD* (*y*, *z*, *x*) is the shortest path between *y* and *z* of the section passing through *x* within the radius *R*; *W* (*z*) is the weight of *z*; *P* (*z*) is the ratio of *z* within radius *R*; Total weight (*y*) is the total weight of all the road sections starting from *y* within the radius; and *TPBt* (*x*) is the intermediary centrality.

#### 2.3.2. Kernel Density Estimation Analysis

Kernel density estimation (KDE) is an important method used to identify the urban agglomeration center area [36]. It analyzes the agglomeration areas of different element categories by continuously simulating the distribution density of point elements or line elements within a certain range of grid cells [37]. The density expansion value and the distance between the elements and the density core are negatively correlated [38,39]; that is, the center strength decreases with the distance. We mainly analyzed the kernel density of the POI infrastructure data of six functional categories, including commerce, transportation, public services, residence, green space and industry, in the seven municipal districts of Qingdao. In this way, we could analyze the agglomeration areas of different functions. We assumed that *x*1, *x*2, …, *x*n are the sample points under study, and their probability density function is *f* (*x*) [40]. Thus, the kernel density estimation expression is as follows:(3)$$f_{n}\left(x\right)=\frac{1}{n\pi r^{2}}\sum_{i=1}^{n}K{\left[\left(1-\frac{{\left(x-x_{i}\right)}^{2}+{\left(y-y_{i}\right)}^{2}}{r^{2}}\right)\right]}^{2}$$
where *K* [ ] is a kernel function; (*x* − *x_i_*)^2^ + (*y* − *y_i_*)^2^ is the square of the Euclidean distance from point (*x_i_*, *y_i_*) to point (*x*, *y*); *r* is the kernel density bandwidth, that is, the search radius; and *n* is the number of points within the radius [41]. The selection of the search radius has an impact on the analysis results. In this paper, the “rule of thumb” is used to calculate the optimal radius *r*. The calculation formula is as follows:(4)$$H=0.9\times \min(\sqrt{\frac{1}{\ln2}}\times D_{m},\;\mathrm{SD})\times n^{-0.2}$$
where n is the total number of elements; D_m_ is the median distance; and SD is the standard distance.

#### 2.3.3. Entropy Methody

The multiple indicator comprehensive evaluation method usually adopts two types: objective evaluation method and subjective evaluation method [42]. The subjective assignment methods, such as the Delphi method and expert scoring method, are usually speculative and random. Therefore, in order to ensure that we obtained objective and reasonable results, we used the entropy method to determine the weight of the indicators [43]. The calculation of the entropy method only depends on the discreteness of the data themselves. The dispersion of indicators is proportional to the amount of information provided, to the impact of this indicator in the comprehensive evaluation, and to the weight. In this paper, the entropy method was used to calculate the weight of the five indices, and the following comprehensive weighted superposition analysis was carried out. In Shannon’s information entropy theory, the information entropy of a group of data is defined as $E_{j}=\ln{\left(n\right)}^{-1}\sum_{i=1}^{n}x_{ij}\ln x_{ij}$. The calculation formula of its weight is as follows:(5)$$D_{i}=\frac{1-E_{j}}{k-\sum E_{j}}\;i=1,\;2,\;\ldots \ldots ,\;k$$
where *D_i_* is the result of index weight calculation; *E_j_* is the information entropy of the indicator data; and *X_ij_* is the standardized value of the original data. If the indicator is positive, $x_{ij}=\frac{e_{ij}-\min e_{ij}}{\max e_{ij}-\min e_{ij}}$. If the indicator is negative, $x_{ij}=\frac{\max e_{ij}-e_{ij}}{\max e_{ij}-\min e_{ij}}$. In these formulae, *e_ij_*, min *e_ij_* and max *e_ij_* are the original index data, the minimum index value and the maximum index value, respectively.

## 3. Results and Analysis

### 3.1. Construction and Analysis of Population Density Index and Night Light Index

The population density grid data of 2020 and NPP-VIIRS night light data of 2022 for Qingdao were, respectively, processed by mask extraction and grid point conversion in ArcGIS. Through inspection, no missing population data were found; thus, spatial interpolation was not required. We directly connected the population density data and night light data to the fishnet in ArcGIS [44]. In this way, the daily and night population density values for each spatial cell within the research scope could be obtained. They could be used as direct indicators for exploring the spatial agglomeration risk [45] (Figure 6).

The results were divided into five grades according to the geometric interval grading method. As can be seen from Figure 6, the population distribution of the seven municipal districts in Qingdao showed an agglomeration trend of “high in the center and low in the periphery” around the areas located along the two sides of Jiaozhou Bay. The population density on both sides is low, and the overall distribution is linear from “northeast to southwest”. At the same time, it was found that the area with the highest density value is probably composed of one main central node and four secondary central nodes. The main central node area includes the whole Shinan District and Shibei District, the south of Licang District and the southwest of Laoshan District. The four secondary central nodes are the north of Licang District and the centre of Chengyang District, the southern area of Jimo District, the coastal area along the northeast side of the West Coast New Area and the central coastal area. At the same time, it was found that the area with the highest density value is probably composed of one main central node and four secondary central nodes. The main central node area includes the whole Shinan District and Shibei District, the south of Licang District and the southwest of Laoshan District. The four secondary central nodes are the north of Licang District and the centre of Chengyang District, the southern area of Jimo District, the coastal area along the northeast side of the West Coast New Area and the central coastal area. According to the policies and regulations of the Land and Space Planning of Qingdao (2021–2035), it was found that the five core nodes are the five core areas of the new urban development pattern of Qingdao: Dong’an City, Chengyang, Huangdao, Jiaonan and Jimo. This showed that the future development center of the city is generally a densely populated area and is also the area at the greatest risk of epidemic disasters. These regions are more likely to cause spatiotemporal concomitant phenomena related to respiratory infectious diseases, such as COVID-19 and influenza, and contact infectious diseases, such as schistosomiasis and other contagious diseases.

The night light indicator reflects the area where people gather at night. It is usually a place that people occupy at night, with a large traffic flow at night and commercial services such as catering, shopping, accommodation, entertainment, etc. These areas are prone to contact transmission, respiratory tract transmission and sexually transmitted diseases such as AIDS and syphilis. Such diseases more likely to spread within the family.

Meanwhile, as an international port city and a central coastal tourist resort city in China, Qingdao has a high population distribution area characterized by important coastal and adjacent ports and docks. Therefore, the important foreign trade port terminals and their surrounding areas should be regarded as high-risk areas for epidemic disasters. Some externally imported infectious diseases can easily break out here. At the same time, infectious diseases such as hand, foot and mouth disease, cholera, viral hepatitis and other digestive tract infections can also easily spread through water, seafood and other foods. Therefore, prevention and control management during the epidemic should be strengthened in these areas.

### 3.2. Construction and Analysis of Road Network Accessibility Indicators

In ArcGIS, we analyzed the SDNA accessibility of the OSM road network in the seven municipal districts of Qingdao. The research adopted a hybrid measurement method combining the angular distance and metric distance, and we selected the infinite distance (N) as the search radius for analyzing the global characteristics of the road network. In this way, we were able to obtain a visual model of the closeness and the betweenness [46] (Figure 7).

Closeness usually reflects the accessibility of the road. It is proportional to the sum of its paths to all other roads. A location with higher accessibility indicates a central location [47]. The roads are shown in red or blue. It can clearly be seen from the closeness analysis chart that the road with the highest accessibility among the seven municipal districts of Qingdao shows obvious centrality. It is basically located in Shinan District, Shibei District, Licang District, the central area of Chengyang District, the southwest of Laoshan District and the south of Jimo District on the north side of Jiaozhou Bay. This location is basically consistent with the downtown area of Qingdao. The flow of traffic and people in this area is significant; thus, it is more likely to cause the occurrence and spread of infectious diseases via respiratory tract transmission and contact transmission.

The Betweenness index belongs to the flow model. It reflects the potential for vehicle or pedestrian crossing movement. The betweenness of roads is proportional to the number of times that people choose to cross the road and is proportional to the potential for traffic flow [48]. The roads are shown or in blue. From the analysis chart of the betweenness of roads, we can clearly see the distribution of roads with a large traffic flow and high travel potential in Qingdao. They are basically located in the main trunk of the roads in the city, the roads along the coast, the main highways and overpasses connecting the surrounding cities and the cross-bay bridges connecting the two banks. These roads can easily carry the flow of people and become the main traffic area for people’s daily commuting and external activities. Compared with other roads, they have a higher concentration risk intensity. Moreover, these roads are highly mobile and belong to the main public space in the city, and they can very easily become an important place for air transmission and easily cause the accompanying phenomenon of time and space. Therefore, the accessibility analysis of the road network can be used as an important indicator of the risk intensity of epidemic disaster agglomeration.

We connected the closeness and betweenness data obtained from the analysis to the fishing net through the ArcGIS space. In this way, the values of the center accessibility and potential for the traffic flow passing through each spatial cell within the research scope could be obtained, respectively. This information could be used as the road accessibility indicator of the risk intensity of spatial agglomeration (Figure 8). The index grid mapping was divided into five grades according to the geometric interval grading method, and the color division was consistent with the linear element mapping.

### 3.3. Construction and Analysis of Functional Mixed Nuclear Density Index

#### 3.3.1. Classification and Analysis of Infrastructure Nuclear Density

We conducted nuclear density analysis on the POI data according to functional categories. According to the “Silverman rule of thumb”, we could determine that the search radius of this nuclear density analysis is 3781.70 m [49]. Through the nuclear density analysis tool in ArcGIS, we could generate a smooth contour surface of the density of interest points. It reflects the spatial agglomeration characteristics of the six functional elements. The analysis results were divided into five grades based on the geometric interval grading method [50] (Figure 9).

It can be seen from Figure 9 that the areas with a high nuclear density of the six types of functional facilities are more concentrated in the spatial distribution. This presented a clear form of “overall gathering, multi-center distribution and circle spread”. However, the concentration area and coverage are not completely consistent. Shibei District and Shinan District are the centers of the high-value core areas in the overall cluster. This showed that the central area is the central urban area of Qingdao with complete infrastructure and a high functional mix, which has driven the development of surrounding areas. At the same time, this area is also the area with the highest risk level for epidemic disasters. From the analysis of the different functional categories, we can observe that the commercial nuclear density mainly includes the distribution of functional sites such as catering, shopping, entertainment, accommodation, daily leisure, etc. Therefore, the area with a high nuclear density can reflect the high-risk areas for respiratory tract transmission, contact transmission, digestive tract transmission (catering sites) and sexually transmitted infectious diseases (accommodation sites). The public service nuclear density mainly includes the distribution of hospitals, schools, governments, stadiums, etc. Thus, the area with a high nuclear density can reflect the high-risk areas for respiratory tract transmission, contact transmission, blood transmission of diseases such as malaria and hepatitis B and mother-to-child transmission of infectious diseases such as HIV (medical places). The areas with a high nuclear density in the functional sites of transportation and industry can reflect the high-risk areas for respiratory and contact infectious diseases. The high nuclear density of residential functional areas can reflect the high-risk areas for respiratory tract, contact and digestive tract infectious diseases. The high nuclear density of green space functional sites can reflect the high-risk areas for respiratory tract transmission, contact transmission and insect-borne infectious diseases such as malaria and kala-azar.

We connected the nuclear density analysis data of the six types of functional POI infrastructure to the fishnet in ArcGIS [51]. In this way, we could obtain the kernel density estimates of each spatial cell within the research scope. This information could then be used as the functional business type classification indicator of the spatial agglomeration risk intensity (Figure 10). The indicator grid mapping was divided into five grades by the geometric interval grading method. The color depth is directly proportional to the concentration density and disaster risk intensity.

#### 3.3.2. Construction of Functional Mixed Nuclear Density Index

In order to build a visual indicator for nuclear density analysis with mixed functions in order to show the spatial distribution of the overall vitality of the city, it is necessary to integrate the nuclear density analysis results of the six categories of functional POI data. Due to the different proportions and degrees of influence of various POIs, this method is not suitable for direct mixed analysis. We must apply certain weights to the six categories of elements [52] and carry out a weighted superposition analysis of the spatial distribution density. Finally, this would enable us to obtain a comprehensive form of infrastructure agglomeration. For this paper, we used the public dependence index to assign weights to the six categories of infrastructure [53]. This index can reflect the attractiveness of various infrastructures to the population and takes the form of a questionnaire in order to obtain data. According to their own cognition, the respondents selected for this study evaluated their dependence on 13 types of POI data according to the scoring method of the Likert 5 subscale. The evaluation criteria are divided into five levels: “unimportant”, “somewhat important”, “generally important”, “relatively important” and “very important”. They correspond to five values ranging from 1 to 5 [54]. In this survey, we collected 204 valid questionnaires, with an effective rate of 94.44%, reliability of 0.925 and validity of 0.918. This proved that the survey data are authentic. The gender proportion of the respondents was equal, the age range was evenly distributed between 18 and 60 years old, the types of occupations were relatively comprehensive, and most of the respondents had lived in the city for more than 5 years. We used SPSS statistical analysis software to calculate the mean value and standard deviation of the obtained data and normalized the mean value of 13 categories. Finally, we obtained the public dependence index of Qingdao’s POI data and the weights of the six functional categories of data [55].

It can be seen from Table 4 that the city public has the highest demands for traffic functional infrastructure, followed by commercial functional infrastructure. The residential function is only used as a rest place, which has little attraction to the public. Therefore, the proportion is relatively low. It can be seen that transportation and commerce are important driving forces for the development of the city, and the public is most dependent on them. Their distribution areas are high-incidence areas for crowd activities and are also the areas that are most likely to lead to the spread of the epidemic. From the perspective of specific functions, people rely more on catering, shopping, life services, medical facilities, school cultural facilities and transportation facilities. Thus, these places are more likely to cause people to gather. During the outbreak of an epidemic, we should focus on strengthening the centralized prevention and control of the distribution points of these functional facilities. The relevant departments should take strict control measures for catering, shopping and transportation facilities. The distribution of medical infrastructure plays a vital role in the prevention and control of the epidemic. It should be configured at fixed points in all high-risk areas to ensure the full coverage of the service scope in high-risk areas.

According to the weight of each functional category, we conducted a weighted superposition analysis of the six functional POI nuclear density values in each spatial unit grid and then obtained the POI Functional mixed nuclear density index. It can be seen from Figure 11 that the nuclear density range after weighted function mixing is 0–251.52, and the analysis results still show the characteristics of multi-center clustering. Additionally, the high-value areas are mainly concentrated in Shinan District, Shibei District and Licang District. However, there are some differences on a small scale, and the results reflect the general public’s cognitive preferences, living habits and dependence, which are widely accepted. This information can more accurately represent the overall spatial vitality distribution level of the region.

### 3.4. Weighted Superposition Analysis of Risk Indicators

In this paper, the entropy method is used to determine the weights of five risk index factors, and data entry and analysis are conducted in Stata [56] (Table 5). According to the calculation result for the entropy value, it can be found that the highest entropy value is the Closeness index of the roads, accounting for 0.358; the second is the Population density index, accounting for 0.273; the third is the Functional mixed nuclear density index, accounting for 0.237; the fourth is the Night light index, accounting for 0.131; and the lowest entropy value is the Betweenness index of roads, accounting for 0.001. These results show that the roads with the greatest potential for traffic flow are more likely to lead to the risk of epidemic concentration. The main traffic roads in cities are important media for guiding pedestrian flow lines, promoting regional flow, and forming crowd concentration. They are more likely to cause road congestion, induce the occurrence of crowd aggregation, and accelerate the risk of epidemic spread. Areas with a dense population distribution and areas with highly mixed functions are also important factors that can lead to the risk of an epidemic situation [57]. They play the role of locating the main target places for pedestrians and attracting people to gather in local areas. Therefore, they can cause the spread of the epidemic. The Night light index is characteristic of the concentration and distribution of people at night. However, it basically refer to a closed, indoor place with low mobility. Thus, the vitality is not high. It can only play a part in the spread of the epidemic. The Closeness index of roads can only reflect the approximate relation of a road in the city to the central area. For a more accurate crowd gathering place, we cannot obtain it directly. Therefore, it cannot play an important role in forecasting and analyzing the risk intensity of epidemic disasters. Thus, its weight is relatively small.

We used the weight proportion of the five risk index factors obtained by the entropy method to conduct the final weighted superposition analysis of the index values in each spatial unit grid [58]. In this way, we could obtain the final factor-weighted superposition risk quantitative assessment results [59] (Figure 12). These results reflect the most accurate spatial risk level zoning and the spatial structure of comprehensive epidemic disasters. According to the geometric interval grading method, the analysis results were divided into five levels of risk areas. The distribution characteristics of the first-level risk area can reflect the spatial structure of the epidemic disaster in Qingdao. On the whole, the distribution presents the characteristics of “one main area, four secondary areas, one belt and multiple points” and has spatial diffusion. Among these areas, the “one main area” is the main risk concentration center of the city. It covers the whole area of Shinan District, Shibei District, the southeast area of Licang District and the southwest area of Laoshan District, while the “four secondary areas” are the four secondary risk concentration centers in the city. They cover the central area of Chengyang District, the southern area of Jimo District, the northeast coastal area of the West Coast New Area and the central coastal area. “One belt” refers to the coastal risk transmission density zone from the northeast to the southwest, and the “multiple points” refer to other scattered local spatial gathering risks posed by the combination of a high population distribution, high traffic flow and high functions.

### 3.5. Quantitative Assessment and Verification of Space Risk

This paper takes the distribution of historical statistical data of the COVID-19 outbreak that occurred in recent years as an example to test the results of the quantitative assessment of the spatial risk of epidemic disasters [60]. From 21 November to 10 December 2022, during the high-incidence period of the epidemic, we obtained a total of 153 high-risk sites within the seven municipal districts of Qingdao and analyzed the data of the high-risk sites and the results of the quantitative spatial risk assessment. It was found that all the high-risk sites are located in the primary risk area and the secondary risk area, including 139 data sites located in the primary risk area. There are 14 data locations located in the secondary risk area (Figure 13). Table 6 showed the specific distribution of high-risk locations within the seven municipal districts of Qingdao. It can found that most of the high-risk locations are in Shinan District, Shibei District, Licang District and Chengyang District. Additionally, these areas are the main and secondary risk concentration centers in the spatial structure of epidemic disasters. It can be verified that the quantitative assessment results of the spatial risk obtained by weighted superposition of the population density grid data, night light data, road network centrality data, road network traversal data and POI Functional mixed nuclear density data are relatively accurate. They can be used to predict the risk level area and risk scale of epidemic disasters. Additionally, the five indicators, namely, the Population density index, Night light index, Closeness index of roads, Betweenness index of roads and Functional mixed nuclear density index, are indeed risk factors leading to an increased risk of epidemic transmission. Therefore, the prevention and control of interpersonal epidemics should be considered in light of these aspects.

## 4. Discussion

Starting with the social activity dimension and the physical space dimension, we selected five indicators, including the population density, night lights, road network centrality, road network traversal and functional mixed nuclear density, to carry out the quantitative assessment of the spatial risk of epidemic disasters. Taking the seven municipal districts of Qingdao as the research scope, we predicted the regional risk level and spatial structure of various epidemic disasters in city areas that are prone to the outbreak and spread of epidemic diseases [61]. Through the quantitative assessment of the spatial risk of epidemic disaster and the comprehensive analysis of the disaster risk indicators and risk impact degree, we can draw the following conclusions:

① The influence of the spatial risk factors of epidemic disasters. Roads with a large traffic flow are the main factor leading to the risk of epidemic concentration and play a role in providing regional flow. The populated areas and the areas with a large mixture of infrastructure functions are the secondary factor leading to the risk of epidemic gathering and play a role in determining the gathering place of pedestrian traffic. The areas populated at night play a part in the spread of the epidemic. Their regional liquidity is low, and their vitality is not high. The accessible central area of the road network has little effect on the identification of epidemic risk areas and can only reflect the approximate concentration area in the city.

② The analysis results of different functional sites reflect the high-risk areas for epidemic diseases with different transmission properties. The population density distribution can reflect the high-risk areas for respiratory tract and contact-transmitted infectious diseases. The night light distribution can reflect the high-risk areas for respiratory tract, contact-transmitted and sexually transmitted infectious diseases. The coastal port terminal area can reflect the high-risk area for external imported epidemics and digestive tract infectious diseases. The accessible area in the center of the road network can reflect the high-risk area for respiratory tract and contact-transmitted infectious diseases. Roads with a large traffic flow can reflect the high-risk areas for respiratory tract infectious diseases. The distribution of commercial facilities can reflect the high-risk areas for respiratory-tract-transmitted, contact-transmitted, sexually transmitted and digestive-tract-transmitted infectious diseases. The distribution of public service facilities can reflect the high-risk areas for respiratory tract, contact-transmitted, blood-transmitted and mother-to-child-transmitted infectious diseases. The distribution of transportation and industrial facilities can reflect the high-risk areas for respiratory tract and contact-transmitted infectious diseases. The distribution of residential facilities can reflect the high-risk areas for respiratory tract, contact-transmitted and digestive tract infectious diseases. The distribution of green space facilities can reflect the high-risk areas for respiratory tract, contact-transmitted and insect-borne infectious diseases.

③ Key prevention and control points of the urban space. Catering, shopping, life services, hospitals, schools and transportation facilities are more likely to cause people to gather. The government should conduct centralized prevention and control measures at the distribution points of such functional facilities. At the same time, medical facilities should be established at fixed points in all high-risk areas to ensure the full coverage of services.

④ Risk intensity zoning and spatial structure of epidemic disasters. The risk intensity of epidemic disasters was divided into five levels. Among them, the spatial structure of epidemic disasters, composed of the first-level risk areas, is characterized by “one main area, four secondary areas, one belt and multiple points” and has the characteristics of spatial diffusion. “One main area” is the main risk concentration center of the city; the “four secondary areas” are the four secondary risk concentration centers in the city; “one belt” refers to the coastal risk transmission density zone from the northeast to the southwest; and “multiple points” refer to other local spatial concentration risk points.

In recent years, the academic community has made significant progress in exploring the prevention and control of various pandemics from the perspective of time and space behavioral geography. Cao Zhidong et al. selected nine spatial risk factors, such as the population, roads, hospitals, schools and shopping malls, to conduct spatial correlation analysis for the incidence rate of SARS and determined that these factors are likely to cause an epidemic [62]. By exploring the distribution and influence of factors affecting the SARS epidemic in Beijing in 2003, Wang Jinfeng et al. determined that the epidemic was mainly concentrated along the traffic line, with spatial diffusion. At the same time, they found that the geographical location, population and hospital distribution were important factors influencing the spatial spread of the epidemic [63]. This proves that the spatial risk factors selected in this paper are accurate and consistent with the research results of this paper. However, the existing research has basically focused on the analysis of the spatial trajectories of individual cases, studied the impact of virus diffusion caused by population flow through the construction of dynamic models, or predicted the characteristics of epidemic spread through the interactions of social networks [64,65]. The current research has only focused on the prediction of individual cases or interpersonal transmission relationships. The precise prediction of the epidemic situation, urban spatial layout structure and regional risks has not been completely solved [66]. In addition, the urban disaster risk assessment system for epidemic disasters used in resilience construction is not perfect [67].

Therefore, on the basis of previous studies, we did not study the spatial distribution of epidemic cases. Instead, starting with the spatial distribution of these influencing factors, the authors used the research method of combining the spatial and geographical distributions of different influencing factors to build the risk level zoning and the spatial structure of the epidemic disaster. This paper identified high-risk areas for different types of epidemics and, finally, tested the spatial distribution of the epidemic case data to ensure the accuracy of the model analysis results. This method can be used in the absence of case data, drug intervention and policy prevention and control at the beginning of an epidemic. It is practical, operable and accurate. It is of great research value for the prevention and control of all human-to-human infectious diseases and the identification of risk areas. However, the epidemic risk indicators selected in this study are only related to the population and spatial/geographical distribution of urban built-up areas, and the impact factors affecting an epidemic are not limited to those used in this article. Therefore, there are certain limitations. In the future, the authors will study ways to combine epidemic impact factors studied in all fields with spatial geographic analysis and improve the quantitative assessment system of the epidemic disaster spatial risk. In this way, we could continue to promote significant progress in the application of spatial geographic analysis in the field of urban public security prevention and control using spatial big data.

## 5. Conclusions

With the prevalence of various kinds of interpersonal epidemics worldwide, the prevention and control of sudden public health events have become an important problem that needs to be solved. Therefore, the quantitative risk assessment of epidemic disasters in regard to “disaster resilience” needs to be improved. In this context, this paper proposed a new idea: to combine the spatial prediction method of the urban agglomeration risk with the quantitative assessment of the epidemic disaster risk using spatial big data. This study comprehensively assessed and considered the risk factors for epidemic disasters and the scale of the risk impact, assisted the relevant urban planning departments in taking measures for emergency prevention and control, and curbed the further spread of the epidemic from the source in a timely manner. Taking the seven municipal districts of Qingdao as the research scope, we selected population density grid data, night light data, OSM traffic network data and POI infrastructure data for spatial analysis in ArcGIS and constructed the Population density index, Night light index, Closeness index of roads, Betweenness index of roads and Functional mixed nuclear density index, respectively. Finally, we calculated the weights of the five indicators using the entropy method, carried out factor-weighted superposition analysis, and, finally, obtained five epidemic disaster risk grade divisions and a disaster spatial structure. The results are as follows: ① The roads with a large traffic flow are most likely to lead to the risk of urban spatial agglomeration, and the populated areas and areas with a large mixture of infrastructure functions are also important factors causing the risk of epidemic agglomeration. ② The distribution of the population during the day and night, the road network that reaches the central area and the roads with a high passing potential, as well as different functional places, such as those for commerce, public services, transportation, residence, industry and green spaces, can reflect high-risk areas for epidemic diseases with different transmission properties. ③ Catering, shopping, life services, hospitals, schools and transportation functional places are more likely to cause crowd gathering. The government should conduct centralized prevention and control at the distribution points of such functional facilities. At the same time, medical facilities should be established at fixed points in all high-risk areas to ensure the full coverage of services. ④ The risk intensity of epidemic disasters is divided into five risk grade areas. Among them, the spatial structure of epidemic disasters, composed of the first-level risk areas, is characterized by “one main area, four secondary areas, one belt and multiple points” and has the characteristics of spatial diffusion.

In general, the quantitative assessment of the spatial risk of epidemic disasters based on spatial big data can be used to test the risk factors that lead to the spread of epidemic disasters, predict the scope of their aggregation risk levels, and build the spatial structure of epidemic disasters. It is helpful for accurately locating the risk areas and epidemic transmission routes in megacities that are prone to outbreaks or cause epidemic transmission. This could help the relevant personnel to contain the epidemic in a timely manner from the early stage of the spread of the epidemic and take appropriate prevention and control optimization measures for the risk factors of the city. The paper improves the disaster risk assessment system used in the construction of resilient cities, taking the risk assessment of public health events as its focus. However, although the five indicators assessed in this paper can accurately reflect the location and spatial structure of the key risk areas for the epidemic outbreak, they still need to be further deepened and improved. In the future, the authors will continue to study ways to combine more indicators of the socio-economic dimension, physical space dimension and natural meteorological environment dimension with spatial geographical analysis to build a comprehensive and more refined indicator evaluation system, thus rendering the regional prediction and analysis results of the risk grade more accurate. The authors will also continue to study issues such as urban resilience after the outbreak of human-to-human infectious diseases so as to strengthen the construction of resilient cities.

## Figures and Tables

**Figure 1 ijerph-20-03274-f001:**
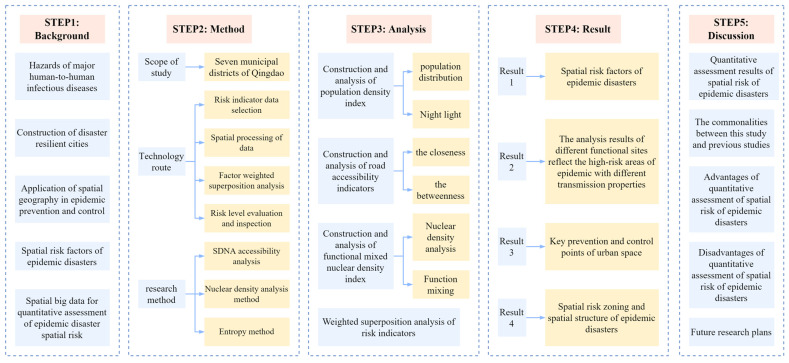
Article framework.

**Figure 2 ijerph-20-03274-f002:**
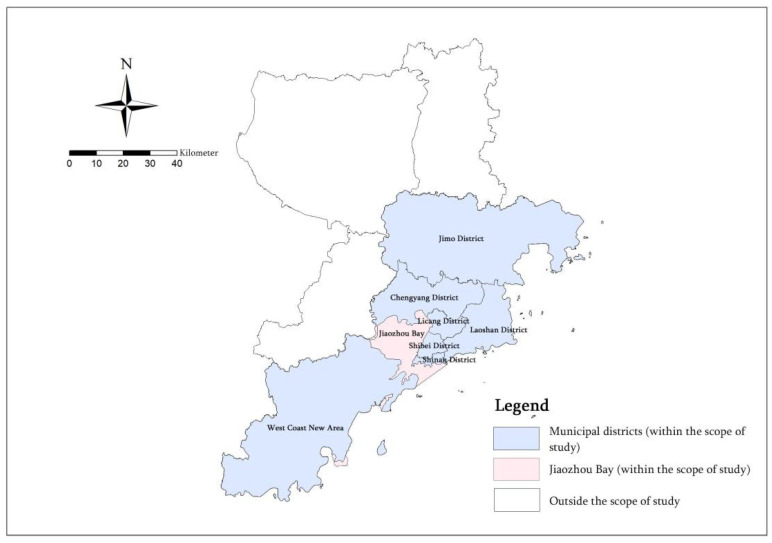
Map of the Study Area.

**Figure 3 ijerph-20-03274-f003:**
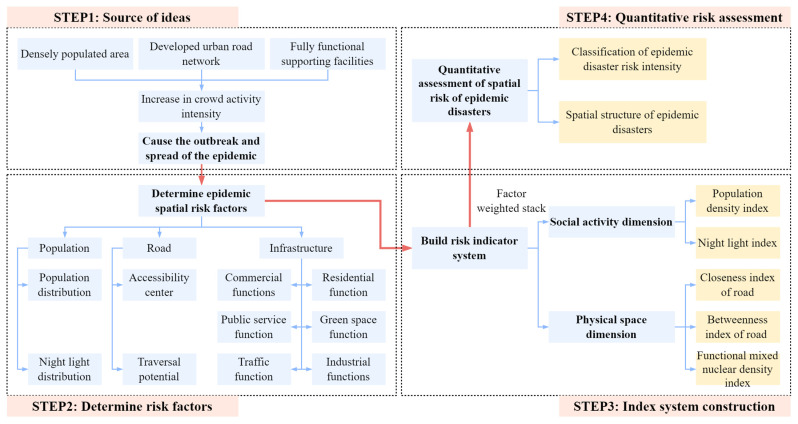
Research objectives and concepts.

**Figure 4 ijerph-20-03274-f004:**
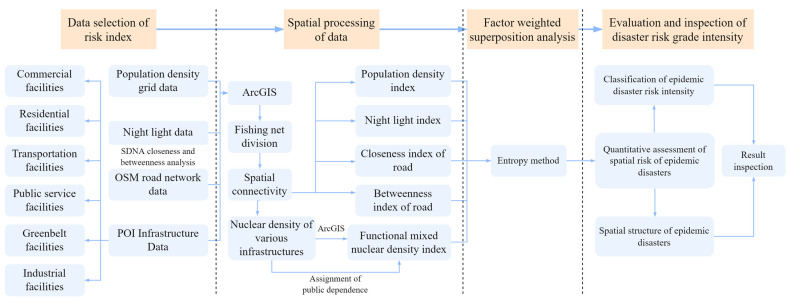
Research technology route.

**Figure 5 ijerph-20-03274-f005:**
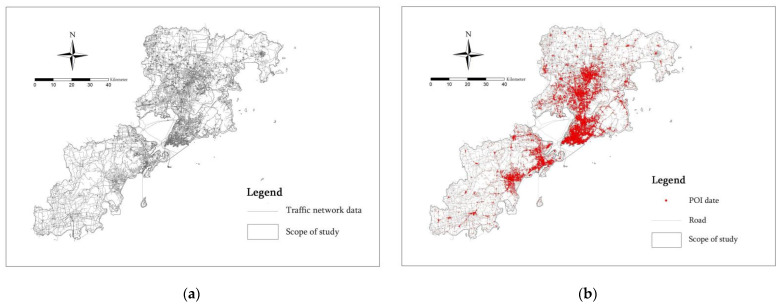
(**a**) Traffic network data and (**b**) POI infrastructure data for the seven municipal districts of Qingdao.

**Figure 6 ijerph-20-03274-f006:**
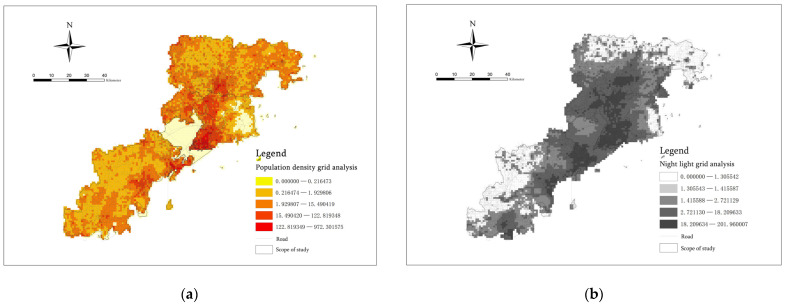
(**a**) Population density index and (**b**) Night light index.

**Figure 7 ijerph-20-03274-f007:**
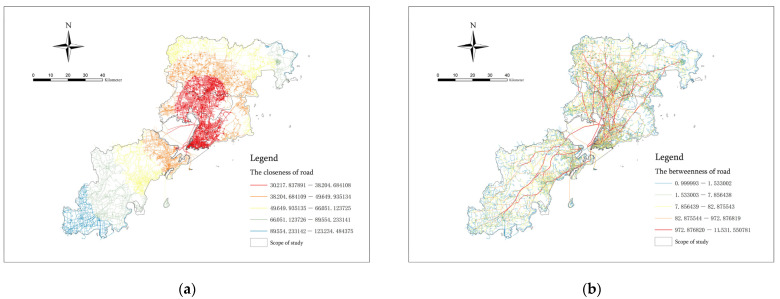
Visual analysis of (**a**) the closeness and (**b**) the betweenness of roads.

**Figure 8 ijerph-20-03274-f008:**
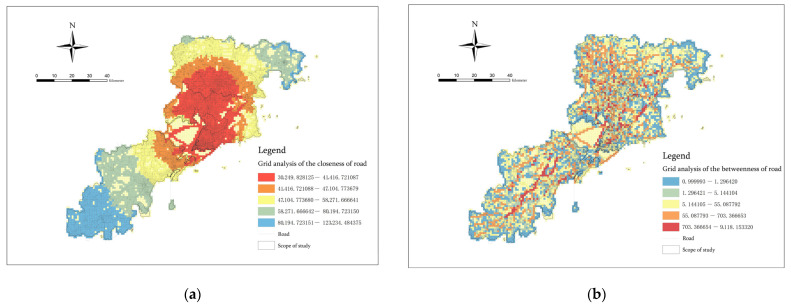
(**a**) The Closeness index of road and (**b**) the Betweenness index of road.

**Figure 9 ijerph-20-03274-f009:**
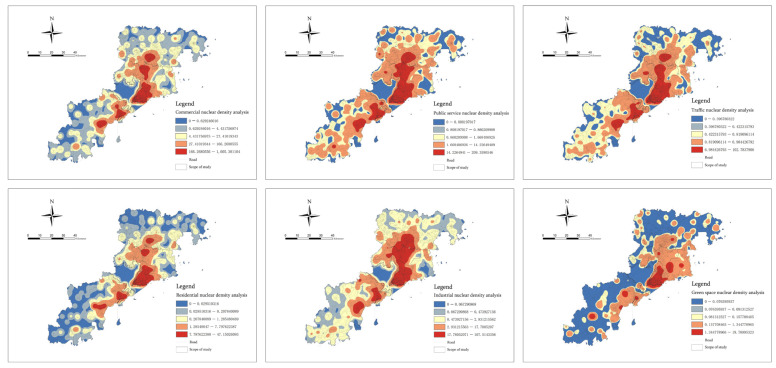
Kernel density estimation analysis of POI data representing six functional categories.

**Figure 10 ijerph-20-03274-f010:**
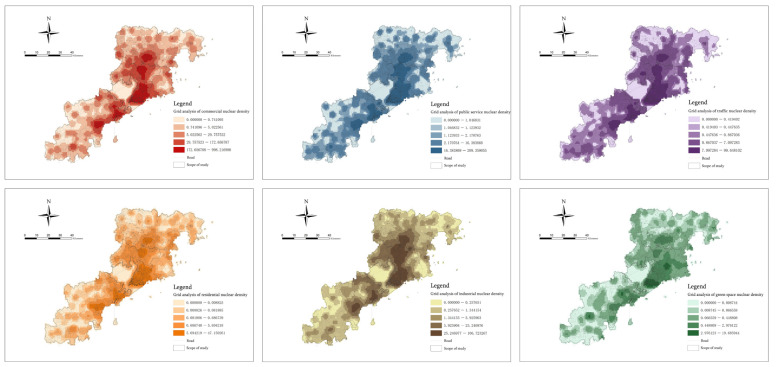
Kernel density estimation analysis of POI data representing six functional categories.

**Figure 11 ijerph-20-03274-f011:**
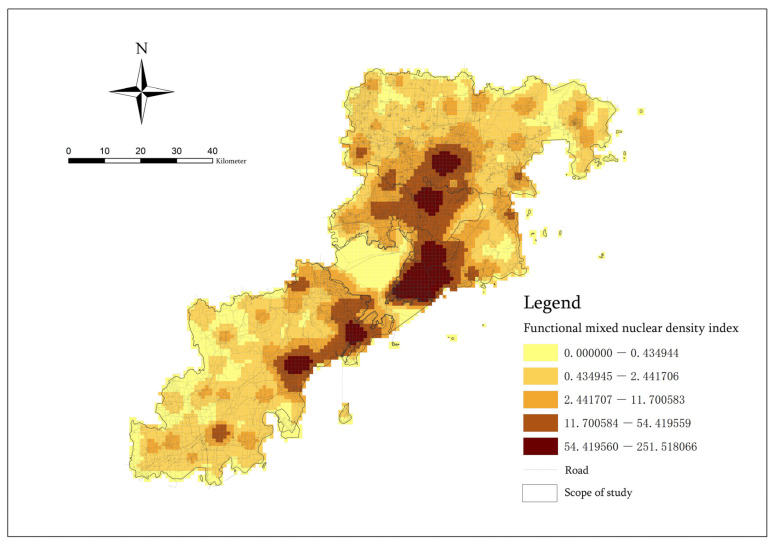
Functional mixed nuclear density index.

**Figure 12 ijerph-20-03274-f012:**
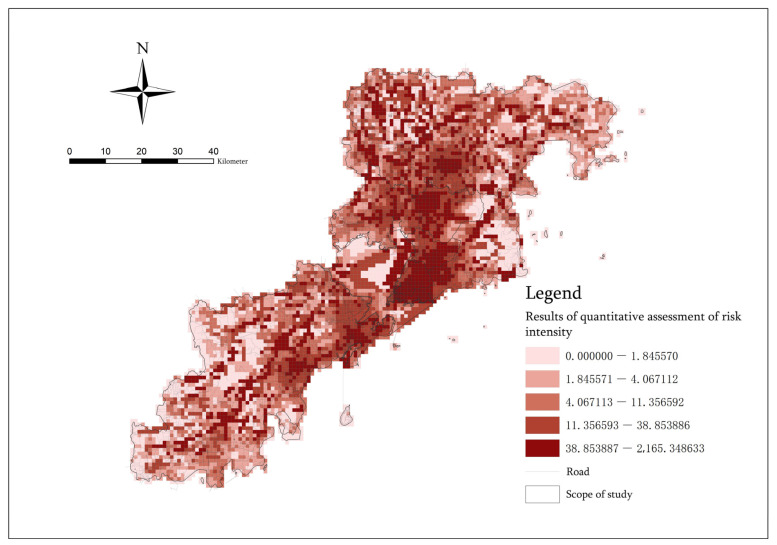
Factor-Weighted Overlay Risk Quantitative Assessment.

**Figure 13 ijerph-20-03274-f013:**
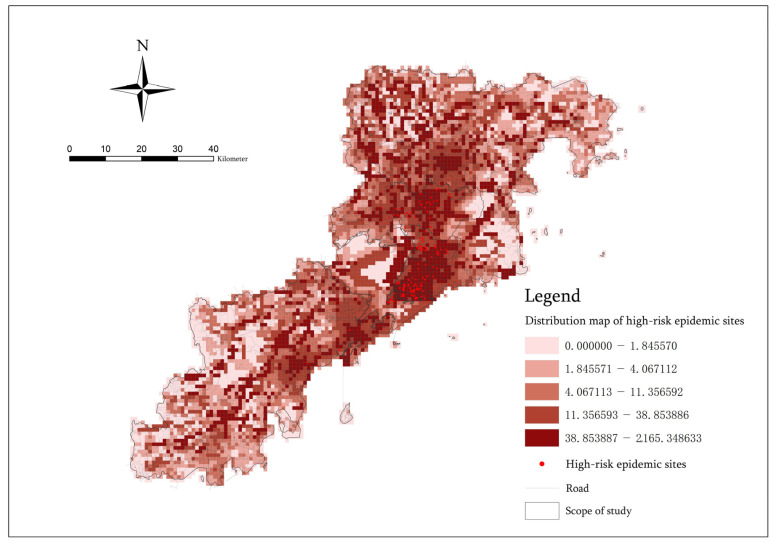
Distribution map of high-risk epidemic sites.

**Table 1 ijerph-20-03274-t001:** Disaster Risk Indicators.

Dimensions	Index (Risk Factors)		Indicator Characteristics	Data Sources
Social activity dimension	Population density index		Daily crowd activity density	Woldpop dataset
	Night light index		Night population distribution density	DMSP-OLS dataset
Physical space dimension	Closeness index of roads		Regional access to central section	OSM open-source network map platform
	Betweenness index of roads		Traffic flow potential	
	Functional mixed nuclear density index	Commercial facilities	Agglomeration density after mixed functions of various infrastructures	Gaode, Baidu POI crawling
		Public service facilities		
		Traffic facilities		
		Green space facilities		
		Residential facilities		
		Industrial facilities		

**Table 2 ijerph-20-03274-t002:** Classification of POI data for seven municipal districts of Qingdao.

Major Categories	Subcategory	POI Quantity	Total POI	Percent %
Commercial function	Catering service	54,304	231,128	65.4%
	Shopping service	92,448		
	Accommodation service	12,364		
	Living service	68,669		
	Leisure and Entertainment	3343		
Residential function	Business residence	9250	9250	2.6%
Public service function	Medical care	14,439	51,439	14.6%
	Education and Culture	17,474		
	Government agencies	15,507		
	Athletic sports	4019		
Traffic function	Transportation facilities	21,379	21,379	6%
Green space function	Scenic spots	2007	2007	0.7%
Industrial function	Companies	37,940	37,940	10.7%

**Table 3 ijerph-20-03274-t003:** Number of areas with high risk of epidemic situation within seven municipal districts of Qingdao.

Area	Chengyang District	Shibei District	Shinan District	West Coast New Area	Licang District	Laoshan District	Jimo District	Total
Quantity	36	38	39	4	32	5	0	154

**Table 4 ijerph-20-03274-t004:** Public dependence on various POI data and proportion of each function weight.

Major Categories	Subcategory	Mean Value	Standard Deviation	Public Dependence	Weight
Commercial function	Catering service	4.588	0.707	0.794	0.175
	Shopping service	4.784	0.518	0.892	
	Accommodation service	3.814	1.164	0.407	
	Living service	4.270	1.003	0.635	
	Leisure and Entertainment	3.574	1.271	0.287	
Residential function	Business residence	3.191	1.082	0.096	0.130
Public service function	Medical care	4.221	0.918	0.610	0.165
	Education and Culture	4.368	0.748	0.684	
	Government agencies	4.093	0.960	0.547	
	Athletic sports	3.240	1.143	0.120	
Traffic function	Transportation facilities	4.961	0.195	1.000	0.200
Green space function	Scenic spots	4.064	0.983	0.532	0.167
Industrial functions	Companies	3.951	1.026	0.475	0.162

**Table 5 ijerph-20-03274-t005:** Weights of the five categories of risk indicators.

Indicators	Population Density Index	Night Light Index	Closeness Index of Roads	Betweenness Index of Roads	Functional Mixed Nuclear Density Index
Weights	0.273	0.131	0.001	0.358	0.237

**Table 6 ijerph-20-03274-t006:** Distribution of high-risk areas in seven municipal districts of Qingdao.

Area	Shinan District	Shibei District	Licang District	Chengyang District	West Coast New District	Laoshan District	Jimo District	Whole Area
Level I risk area	34	34	20	26	3	4	0	121
Level II risk area	5	4	12	10	1	1	0	33

## Data Availability

Not applicable.

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
