# Peer review of "Exploring the Quantitative Assessment of Spatial Risk in Response to Major Epidemic Disasters in Megacities: A Case Study of Qingdao"

_ijerph, 2023, doi:10.3390/ijerph20043274_

Round 1

Reviewer 1 Report

This paper focuses on the quantitative assessment of spatial risk in response to significant epidemic disasters in megacities.

I found the research topic interesting and suitable for publication in the journal. However, the paper must be improved due to several reasons below.

1.       Generally, the sections of the papers are easy to follow by their titles. However, the paper's content is very lengthy. The sections are unorganized and hard to follow (rather boring to read). The manuscript contains various typos and hard-to-understand sentences. Mistaken heading (3 – Results on page 6), repeated headings (for 4.1 and 4.2), and section numbers (2.2.2, 2.2.3, 2.2.4) are found.

 2.       Regarding the literature review and fundamental background: There is a need to present and discuss the spreading nature of major epidemic disasters. Why are spatial perspectives important for these events?

 A separate section on literature review and background knowledge about spatial perspectives of epidemic outbreaks, tied with human behavior, is essential because it helps to justify the chosen data and the modeling approach. Currently, what has been done in this field is only mentioned briefly (lines 55-91), and why epidemic disaster risk should be modeled as it is using the method presented here is not clarified.

 The discussion in the current version is also short regarding the future development of this work. For example, in the second last sentence of the manuscript, the authors mentioned "indicators of other dimensions than physical and social dimensions" but don't explain these dimensions well and have no further explanation on "other." I suggest having a background section presenting all these dimensions, how they can be modeled, what dimensions the paper is trying to accomplish, and its novelty and limitations.

 3.       Regarding Section 2 (Materials and Methods): This section is too long and somewhat unorganized. Figure 1 is helpful for the reader to connect all presented components but was not emphasized and was in the Introduction section. The methods presented in their general forms are ok, but it would be better to connect them with the datasets applied for this study.

 4.       I have the following methodological questions:

 4.1. The papers model risks using the four variables: population density, road centrality (closeness and betweenness), night lights, and functional places. Human epidemic diseases are transmitted via either indirect contact through respiratory or direct contact through saliva, skin and sexual intercourse. How can these functional places be used differently in the model to reflect risks for these different forms of transmission? If considering the forms of transmission, spatial structures of different functional places can be vital for one kind but not the other.

4.2. The finding patterns seem to agree well with the spatial pattern of population and infrastructure (there is a higher risk where high concentration is) – which is not useful. Can the model provide any further valuable insights? For example, considering the different natures of functional places helps with different kinds of outbreaks and potentially provides more insight and useful results.

4.3. How do we count for the spatial choice behavior, i.e., people's flows to work and entertainment resources (eat, shop, social )? – again, going back to the importance of considering particular functional places

4.4. Is the model result validated by the COVID data in any way? The authors mentioned the data, but I didn't see the validation part presented.

Author Response

Dear reviewer: Thank you for your decision and constructive comments on my manuscript.We have carefully considered the suggestion of Reviewer and make some changes. We have tried our best to improve and made some changes in the manuscript. According to your modification suggestions, the red part is our reply. Revision notes, point-to-point, please see the attachment.

Reviewer 2 Report

This study conducted a quantitative assessment of the spatial risk of the epidemic that included population density index, night light index, closeness index of road, betweenness index of road, functional mixed nuclear density index so as to obtain the classification and spatial structure of the epidemic risk intensity. This is an important study, but there are still some points that need to be clarified and verified.

1  In the article, the population density is divided into six grades according to the geometric interval classification method, the closeness index of road and the betweenness index of road was divided into six grades using the natural breakpoint method. Why they were used different method to divided into six grades instead, what is the using standard?

Finally, risk epidemic sites were divided into risk areas of 1-5 levels. However, the impacts including in the research (e.g. population density, night light index, closeness and the betweenness index of road) were divided into six. Why the epidemic risk is not divided into the same levels as the factors.

3  In the section of 4.1, verification of epidemic risk was included that belonged to research results and was not suitable for discussion section.

4  Line384-385: Based on the population density grid analysis and night light grid analysis, the area was divided into the first, second and third level risk characterization level areas according to the density value. Please clarify how can the continuous value of nuclear density be divided into three risk levels.

5  There are many factors affecting the epidemic, such as population density, social interaction, personal protection measures, etc. This study uses five factors to quantify the spatial risk of the epidemic, but does not consider protection measures and the impact of external input risk. In addition, this study did not quantitatively analyze the relationship between risk factors (such as population density) and epidemic situation. These deficiencies should be pointed out.

6  Line 165-166: Thus, this study uses the grid data of population density from 2000 to 2020 provided by Worldpop. Please clarify whether this is the population density of a certain year or the average of many years.

Author Response

(The authors gave the same response as above.)

Reviewer 3 Report

In the article, the authors describe the method of determining various levels of epidemic risk based on the population density index, night light index, closeness index of road, betweenness index of road and Functional mixed nuclear density index. The article is preceded by an in-depth review of the literature on the subject.

The article has only minor inaccuracies and elements that in my opinion require improvement:

- in many drawings the font with descriptions and legend is too small

- line 107, 109 ... there is an incorrect term in 9 places

"fishing nets in ArcGIS" > fishnet?

- the method chapter lacks a description of where the spatial data was obtained from

Author Response

(The authors gave the same response as above.)

Round 2

Reviewer 1 Report

Thank you for addressing my comments. I do see that the paper is now improved.

Reviewer 2 Report

No further comment to authors.